# "It left me burnt": Traditional treatment and stigma experiences of cutaneous leishmaniasis in Kalu district, Ethiopia

Teklu Cherkose[1]*, Kibur Engdawork[2], Maria Zuurmond[3], Abebaw Y. Alemu[1,4], Fikregabrail Aberra Kassa[1], Saba Lambert[5], Zenebu Begna[1,6], Katherine Halliday[7], Tara Mtuy[8], Yohannes Hailemichael[1], Michael Marks[5,9], Mirgissa Kaba[10], Endalamaw Gadisa[1], Stephen L. Walker[5], Jennifer Palmer[8], SHARP collaboration[¶]

1 Armauer Hansen Research Institute, Addis Ababa, Ethiopia, 2 Department of Sociology, Addis Ababa University, Addis Ababa, Ethiopia, 3 Department of Non-Communicable Disease Epidemiology, London School of Hygiene & Tropical Medicine, London, United Kingdom, 4 Department of Epidemiology and Biostatistics, University of Gondar, Gondar, Ethiopia, 5 Department of Clinical Research, London School of Hygiene & Tropical Medicine, London, United Kingdom, 6 Department of Public Health, College of Medicine and Health Sciences, Ambo University, Ambo, Ethiopia, 7 Department of Disease Control, London School of Hygiene & Tropical Medicine, London, United Kingdom, 8 Department of Global Health and Development, London School of Hygiene & Tropical Medicine, London, United Kingdom, 9 University College London, London, United Kingdom, 10 School of Public Health, Addis Ababa University, Addis Ababa, Ethiopia

¶ Skin Health Africa Research Programme is provided in the Acknowledgements
* chteklu@gmail.com

## Abstract

### Background

Cutaneous leishmaniasis (CL) is a major public health issue in Ethiopia, often causing lesions on the cheeks, nose, and lips that take months to heal, and leave permanent scars. Data on the lived experiences of people with CL in Ethiopia is needed to support design of interventions that could respond to their needs.

### Methods

We interviewed 18 people with active or healed CL to understand their experiences of the disease in Kalu district, South Wollo, Ethiopia. Interviews were audio recorded, transcribed and translated for content analysis which focused on experiences of symptoms, treatment and the consequences of living with CL.

### Results

Experiences of CL symptoms and treatments used by people with CL in Kalu were associated with physical discomfort and significant emotional distress. CL began with pain, pruritus, bleeding and ulceration. Most people with CL used painful treatments such as plant-based traditional medicines which irritated the skin or heat to cauterize lesions at home or by traditional healers and expressed dissatisfaction at being left

**Data availability statement:** The data underlying this study contain potentially identifiable and sensitive information from research participants. Due to ethical obligations to protect participant confidentiality, as approved by the relevant Ethics Committees, and in accordance with national data protection regulations, these data cannot be made publicly available. De-identified data may be made available to qualified researchers upon reasonable request, subject to review and approval by the institutional Ethics Committee to ensure that the proposed use is consistent with the original ethical approvals and participant consent. Requests for access to the data should be directed to: AHRI -ALERT ETHICS REVIEW COMMITTEE (AAERC) ahri.alerterc@ahri.gov.et P.O. Box 1005 Armauer Hansen Research Institute (AHRI), Addis Ababa, Ethiopia.

**Funding:** This research was carried out as part of the Skin Health Africa Research Programme (SHARP) funded by the National Institute for Health and Care Research (NIHR https://www.fundingawards.nihr.ac.uk/award/NIHR200125) (NIHR200125) using the Research and Innovation for Global Health Transformation (RIGHT) Programme (SLW, JP, EG, MM, MK, SL). The views expressed in this article are those of the authors, not necessarily those of the NIHR. The funders had no role in study design, data collection and analysis, decision to publish, or preparation of the manuscript.

**Competing interests:** The authors have declared that no competing interests exist.

"burnt" but not healed. During treatment, individuals reported abstaining from sexual intercourse as this was believed to worsen CL; people also avoided contact with others who'd recently had sex. Individuals with CL experienced psychological distress, reduced self-worth, and self-exclusion from social participation due to anticipated or experienced stigma, fear of spreading the disease and worsening their own disease.

## Conclusions

Community engagement strategies to promote early case detection and treatment at health facilities should acknowledge the specific fears, informational needs and challenges that shape existing care-related behaviours of people with CL. Providing information about the safety of common traditional treatments and correct information about contagiousness are key areas that public health programmes could address to reduce some of the disease's impacts. Existing cultural attitudes that emphasize shared vulnerability to CL and underlie non-stigmatizing behaviors could also inform stigma interventions.

## Author summary

Ethiopia is significantly affected by all forms of leishmaniasis. Cutaneous leishmaniasis (CL) is a significant public health problem with an estimated incidence of 50 000 annually. The skin lesions mostly appear on the face. The visible changes caused by the disease have considerable implications on the psychosocial wellbeing of affected people. This study aimed to understand how people affected by CL make sense of signs and symptoms, locally available treatments and the consequences of living with the disease. People affected by CL in Kalu, Ethiopia experienced physical and emotional discomfort in relation to signs and symptoms and traditional treatments applied. Change in appearance, fear, sadness, anxiety, stigma and abstinence from sexual intercourse were reported as consequences of living with CL. To reduce some of the disease's impacts, our findings imply the need for interventions to target key cultural beliefs that inform needs and behavior of the affected people.

## Introduction

Cutaneous leishmaniasis (CL) is a neglected tropical disease (NTD) of the skin, caused by protozoa of the genus *Leishmania* [1] and transmitted by sandflies. CL is characterised by ulcers, nodules, and plaques following the bite of infected sandflies [2]. Skin lesions occur mainly on the face [1] and are often refractory to treatment [1]. CL results in permanent damage due to scars which may be associated with marked stigma [3–5]. Ethiopia has one of the highest burdens of CL with an estimated annual incidence of 50 000 cases [3,6] with children and young adults at most risk [2,7]. *L. aethiopica* is responsible for nearly all CL in Ethiopia [8–10].

CL is associated with reduced health-related quality of life in many settings including Ethiopia [11–13]. Scarring and damage to exposed areas of the body contribute to psychological distress and feelings of fear, regret, anger, shame, sadness and sometimes suicidal ideation [14–18]. The psychological effects of CL are often worsened by affected people blaming themselves for the condition [14]. CL affected people experience stigma typified by rejection, isolation, and exclusion associated with active CL lesions and scars [14–17,19]. The early manifestations of CL may not be stigmatizing as they resemble common non-harmful skin conditions such as insect bites and may not trigger care seeking behaviours [4,20]. In societies that prioritize health and attractiveness, losing these traits can lead to profound social and psychological consequences, as deviation from beauty norms often result in rejection and diminished self-worth [21–25]. Individuals may feel compelled by societal pressures to regain health and beauty norms through different treatment modalities [25,26].

Studies on experiences of people with CL in limited resource settings show high reliance on community-based treatment approaches due to socio-cultural, economic, or accessibility issues [7,19,27–31]. Community-based treatment approaches, such as plants, chemicals, heat cauterization, or a combination of these have been reported to contribute to delayed care seeking behavior and are associated with complications [7,28,29,32].

Previous studies on lived experiences of people with CL have emphasized the mental and social impacts of living with CL [15,33,34] but there remains a significant gap in research focused on how these experiences emerge from specific cultural and treatment contexts to help inform the design of appropriate services. The key research question to this study was how treatment and stigma experiences intersect in a specific socio-cultural context, informing experiences of affected people.

Studying the intersection between treatment and stigma experiences of individuals with CL is essential for improving health outcomes and informing public health interventions. Stigma can act as a significant barrier to seeking timely medical care, leading to delayed treatment and exacerbated health issues [4,5,17]. Understanding how stigma affects affected individuals allows for the development of targeted educational campaigns to combat misconceptions about the disease, as well as the creation of holistic treatment approaches that address both physical and psychological needs [5,17]. By fostering community support and integrating mental health resources into care, healthcare providers can enhance engagement and overall quality of life for those affected by CL [35,36].

As part of a larger study dedicated to improving experiences of care for NTDs of the skin [37], we explored the lived experiences of people with CL in terms of how they experience CL signs and symptoms, treatment and social consequences of living with the condition in Kalu, Ethiopia.

## Methods

### Ethics statement

This study was conducted in accordance with the declaration of Helsinki with observance of ethical principles. Ethical approval was obtained from Armauer Hansen Research Institute and All Africa Leprosy, Tuberculosis and Rehabilitation Training Centre (AHRI/ALERT) Ethics Review Committee (PO/26/20), the Ethiopian National Ethical Review Committee (712–506/m259/35) and the London School of Hygiene and Tropical Medicine (22604). Participants were provided with detailed information about the study and formal written consent was obtained. Where necessary the researcher read the consent to prospective participants, and they confirmed consent using a thumb print in the presence of a witness. Adults provided written informed consent to participate for themselves and for children aged under 18 years. Children aged 12–17 years also provided verbal assent.

### Study setting

This study was carried out in the South Wollo zone of Amhara region, in Kalu district, which had a projected population of 250,837 in 2024 [38]. Kalu is endemic for CL and consists of highland and lowland geographic terrain with residents' livelihoods mainly based on subsistence agriculture and livestock rearing. There is no hospital in the district. Primary care is provided through nine government-run health centres, which have limited capacity for inpatient care, and thirty-five health

posts staffed by health extension workers, offering essential health promotion and prevention services but limited treatment. At the time of this study, the nearest place CL could be managed was the dermatology department at Boru Meda general hospital, about 50km from Kalu.

**Participant recruitment**

Fifteen adults (≥18 years) and three adolescents (13,14 & 17 years), self-identified or community-recognized for having active or healed lesions consistent with CL, were recruited through community health extension workers. This purposive sample included people with presumed (n = 17) or confirmed (n = 1) CL, with attention to ensuring balanced representation by gender and age. Participants were recruited using purposive sampling, supplemented by snowball referrals when appropriate, to identify individuals with lived experiences relevant to cutaneous leishmaniasis in the community. Given the limited access to parasitological confirmation in Kalu district, our focus was on how individuals experience and interpret their illness as understood locally to be CL, rather than on clinical classification. No eligible individual declined participation after being approached.

**Data collection**

Data collectors (TC, AYA, YHM) conducted semi-structured interviews during March-May 2021 following a guide designed to collect data on a variety of topics for our larger study including experiences of CL signs and symptoms, care-seeking logics, treatment experiences and economic and social consequences of living with CL, including stigma. The guide and probes were developed through iterative refinement. Participants were interviewed in their home for up to 60 minutes. All interviews were conducted in Amharic and audio-recorded with complementary observation notes made by the interviewers.

**Data management and analysis**

Interview recordings were transcribed and translated into English and translations were quality checked by experienced researchers (KE and ZB) before detailed coding using MAXQDA software [39]. Narrative coding technique was employed to construct stories based on participants' lived experiences. We applied a reflexive thematic analysis approach in which the primary analyst engaged deeply with the data through iterative coding, memo writing, and peer debriefing, emphasizing interpretive reflexivity as the foundation of analytic rigor. Themes were interpreted whilst maintaining participants' perspectives and contextual specificity using researcher self-reflexivity and checking for alignment of findings during analysis.

In this paper, we present themes that emerged from the data in narrative coding by the first author (TC). The themes include CL-affected people's experiences of CL signs and symptoms, treatment and social consequences of living with the disease, including stigma. We sought to classify stigma experiences according to whether they were 'internalized', 'social' or 'structural' [40,41]. Internalized stigma (also referred to as 'self-' or 'felt' stigma) is characterised by negative feelings about self, maladaptive behavior and stereotype endorsement resulting from an individual's actual experience or anticipation of negative social reactions. Social stigma (also referred to as 'enacted' or 'public' stigma) operates at a group level and can result in discrimination. We could not find examples of structural stigma that relates to people being treated inequitably because of unfair rules, policies and procedures within organizations and society. Summary of themes and number of participants contributing to each theme is shown in Table 1. Data from the interviews on other topics are presented in an earlier publication [37].

## Results

**Sociodemographic characteristics of participants**

Thirteen of the 18 people interviewed were male. Three participants were adolescents aged 13, 14 and 17 years and the remainder were above 18 years old; children under 13 were excluded due to concerns with training of data collectors and

**Table 1. Summary of themes and number of participants contributing to each theme.**

| Main Theme | Subthemes | Number of Participants Contributing (n) |
|---|---|---|
| Experiences of CL signs and symptoms | Itching | 8 |
| | Swelling | 12 |
| | Pain | 5 |
| | Ulceration | 16 |
| Experiences of local treatment | Burning or pain from heat application using leaves or iron | 15 |
| | Topical application of different elements as home remedies | 14 |
| Consequences of living with CL | Effects on appearance | 11 |
| | Common anxieties | 13 |
| | Modified sexual behavior | 7 |
| | Stigma (internalized or social) | 10 |

the interview guides that initially didn't consider the need to approach children differently from adult participants. All participants were rural residents whose livelihoods were based on farming. At the time of interview, all participants had at least one visible lesion or scar on their face; eight participants had signs that mainly involved the nose, five the cheek and five participants had multiple facial areas involved. Participant characteristics are shown in Table 2.

## Experiences of CL signs and symptoms

People in Kalu district (including participants) commonly used a vernacular name *Kunchir* or *Konchir* to refer to CL. Interviewees characterized their experiences of early CL as painful, red, swollen, itching, bleeding and ulcerated.

> When it [my illness with CL] started […] it was very painful especially when I bent down [and blood rushed to the area]. […] When it became red, people said it is Kunchir [CL]. (IDI 025, male, 23 years)

An adolescent with CL said

> It [CL] became like this [scabbed] which is still very itchy. When I peel off [the scab], it bleeds (IDI 049, male, 13 years)

> Scarring or nasal anatomical changes were anticipated and feared as late manifestations of CL by most participants.

## Experiences of local treatment

Interviewees reported to have used home remedies (14 of 18) and treatments obtained from traditional healers (15 of 18). Only one participant had experience of allopathic treatment administered at Boru Meda Hospital.

Participants reported various materials that were used to treat CL at home. These included eating honey or applying it to the lesion, topical application of the latex of *Euphorbia abyssinica* or *Euphorbia tirucalli*, heated garlic, and a variety of heated leaves including henna. Many people also cauterized their lesions with hot iron tools. Individuals reported that these treatments were frequently associated with pain, burns and scar formation.

> When someone [a family member] applied the heated leaf on the spot [the ulcerated portion of the lesion], it had a burning sensation. Immediately tears dropped from my eyes. (IDI 023, male, 58 years)

**Table 2. Demographic and clinical characteristics of study participants (N = 18).**

| Participant ID | Age (years) | Gender | Lesion location | Lesion status (active/ healed) |
|---|---|---|---|---|
| IDI 021 | 50 | Male | Leg | Healed |
| IDI 022 | 28 | Male | Face | Healed |
| IDI 023 | 58 | Male | Face | Active |
| IDI 024 | 60 | Male | Face | Active |
| IDI 025 | 23 | Male | Face | Active |
| IDI 026 | 30 | Male | Face | Active |
| IDI 028 | 23 | Female | Face | Active |
| IDI 029 | 46 | Female | Face | Active |
| IDI 038 | 30 | Male | Face | Active |
| IDI 041 | 40 | Male | Hand & face | Active |
| IDI 043 | 30 | Male | Face | Active |
| IDI 046 | 84 | Male | Face | Active |
| IDI 049 | 13 | Male | Face | Active |
| IDI 051 | 30 | Male | Face | Active |
| IDI 053 | 40 | Female | Face | Active |
| IDI 060 | 14 | Female | Face | Active |
| IDI 061 | 27 | Female | Face | Active |
| IDI 070 | 17 | Male | Face | Active |

There were also some individuals who described inflammatory adverse effects of the plants used:

People told me to apply [the heated leaves] for a few seconds. But hoping for fast healing, I applied for a longer time. It hurt me very much. It burned a lot. It turned my eye red since it [the lesion] was close to my eye. (IDI 026, male, 30 years)

As with home application of medicines, participants treated by healers at their sites or following their directions at home described pain and associated anxieties that treatments could be making the lesions worse:

I went to a traditional healer […] He put the nails in the fire for a while and [...] put the hot nail into my body [the ulcerated skin lesion]. I was in agony. (IDI 021, male, 50 years)

The treatment [with the herb] was burning; I felt tingling pain. (IDI 060, female, 14 years)

Reported treatment timelines were often very long (ranging from a few weeks to seven months or more), and the distances people travelled to see healers (up to 40 minutes by three-wheel vehicle or 90 minutes by foot) meant this care-seeking represented a significant investment of time and energy.

I went to a traditional healer who lives around here. He told me that I had Kunchir [CL] and gave me the traditional medicine […] it took 7 months for the wound to heal. (IDI 022, male, 28 years)

As a result of all of these issues, most of the participants who received traditional treatments expressed frustration and disappointment:

It [treatment with a heated iron] was a total waste. It left me burnt. (IDI 021, male, 50 years)

## Consequences of living with CL

The social consequences of living with CL included managing the effects of the disease on appearance, anxieties about disease progression, modified sexual behaviour, and stigma.

### Effects on appearance

Almost all interviewees had CL-like lesions involving the face. The visible changes caused by CL were one of the principal experiences reported by interviewees. Sentiments expressed by participants included:

> It [CL] ruined my appearance. (IDI 053, female, 40 years)

and

> Now my appearance has changed. I don't look like the old me. I look like another person. It takes people a long time to recognize me. …People say to me 'that good-looking face is destroyed'… which affects me psychologically. I feel ashamed. (IDI 023, male, 58 years)

Scars and sometimes disfigurement were viewed as inevitable and permanent manifestations of CL.

> It [CL] is a dangerous disease. It destroys your appearance. For example, it breaks your nose down. …Even if the wound is healed, the scar remains there. (IDI 025, male, 23 years)

Adolescents and young adults (under 30 years) commonly discussed concerns about how CL was likely to have a lasting effect on their appearance. This was also mentioned by all female participants interviewed.

> I feel sad because it [CL lesion on cheek] will create *godanisa* [scar]. (IDI 060, female, 14 years)

### Common anxieties

Individuals with active CL said that they experienced a variety of emotions including sadness, embarrassment and fears about disease progression. The most frequently reported sources of fear were nasal changes, loss of vision, increasing size of facial lesions and even death:

> I am worried that it may affect my eye. You see my teeth… I am worried that it [CL] might affect them too. Right here [nose], I have pain. … It is spreading toward my eyes; it is trying to attack my eyes. I am not sleeping well as I am worried that it might blind me. When it appears in many places it might become cancer. I worry a lot about my situation because I don't have anything [money] to get treatment and I can't do anything about it. (IDI 029, female, 46 years)

### Modified sexual behavior

Adult interviewees reported that they modified their sexual behaviour because they believed that sexual intercourse would exacerbate CL. Seven of the 15 adult participants (5 males and 2 females) revealed that they abstained from sexual intercourse for many months or even years because of this belief.

> I used to have sex before I knew that the illness was *kunchir* [CL]. I think this is the reason why it took a long time [seven months] for the wound to heal. But after I started the treatment, I discussed with my wife and abstained until the wound was healed. (IDI 022, male, 28 years)

In our community, it is believed that it may aggravate the disease and reduce the potency of the traditional medicine. [Therefore], I abstained during application of the traditional medicine and I will keep that way until I totally recovered from this disease. He [my husband] knows the situation very well and he understands me since he knows that this disease harms people resulting in permanent damage to [affected] body parts. (IDI 061, female, 27 years)

I am abstainings from sex for over one year. Nothing is more important than my wellbeing. He [my husband] can leave me if he wants. […] But he hasn't said anything so far. Rather he is worried for me and says 'she was very pretty but this thing [CL] changed her [appearance].' (IDI 029, female, 46 years)

**Stigma**

Some participants (5/18) reported that they felt stigmatized by other people in their community.

If you aren't so good looking, people see you as inferior. I mean when there is some kind of lesion on your face, that is not a nice thing to look at. They [people in the community] are not happy because this thing [CL lesion] is noticeable on my face. (IDI 028, Female, 23 years)

Participants reported being insulted because of their illness.

Yes, it [the word *'kunchiram'*, meaning "you who have CL"] is an insult because people use this to abuse the person, and it is bad. The insult is annoying to the sick person. (IDI 025, male, 23 years)

More than half of the participants (10/18) said that whilst they didn't experience discrimination (social stigma), they internalized stigma feeling shame, inferiority, and embarrassment. Interviewees commonly reported that they regarded themselves as disgusting and inferior because of their CL lesions and this resulted in self-exclusion from social participation.

I think that my face may disgust people who sit and talk with me. It's me… it's just my thought…. Nobody has told me that I look ugly. [For this reason,] for the last four months I isolated from people. (IDI 023, male, 58 years)

Sometimes, it is difficult to let other people see your face when the wound is on the face. You might become ashamed to go outside. (IDI 022, male, 28 years)

Erroneous fears of contagion were also reported to drive stigma and exclusion.

I even use a separate bottle of water not to contaminate my children and my family. I also told my wife to give me coffee with a separate cup. (IDI 022, male, 28 years)

They [people in my village] are not happy because … they think like it [CL] could get transmitted to them. (IDI 028, female, 23 years)

People with active CL lesions were advised to stay at home to avoid interacting with people who had recently had sex (hours or the day before) whose "shadow" or "impure energy" was believed to aggravate the lesion. For this reason, most people with active CL avoid leaving their homes.

[My neighbors] never [discriminate or avoid me]. To the contrary, I am the one avoiding meeting them. It is said mixing with people will cast shadow [unclean energy of persons who had sex in a few hours or days before the meeting] which

aggravates the wound [CL lesion]. I am trying to protect myself from it [shadow] so that it [my CL lesion] will not aggravate or spread to other parts of my body. I should be careful. I don't expect other people to do so [take care]. I am not going [to social gatherings] because I want to avoid the shadow. I am not meeting my neighbors too. (IDI 029, female, 46 years)

Most of the study participants acknowledged that they gained advice, encouragement, and practical support, rather than discrimination from individuals in their community.

Everything is like it was before. It [CL] happens to everyone. People advised me what they know about it [CL]. Some of them told me about the treatment they knew it. We are doing all things together. People say she is sick and let us help her. There is nothing like exclusion from social activities. (IDI 038, male, 30 years)

Interviewees shared their strategies of concealment for coping with anticipated stigma. Many tried to conceal affected body sites including long after the lesions had healed and scarred over to avoid being asked about the cause of the visible skin changes and to make others feel comfortable.

I cover it (CL lesion) because of disfigurement … all people who I meet ask what I have. So as not to embarrass people [make others uncomfortable], it is better to be covered. I could do nothing better. (IDI 041, male, 40 years)

## Discussion

People with CL in Kalu reported significant negative physical and psychological experiences associated with their illness, which persisted even after their skin lesions healed. The effects of CL signs and symptoms and traditional treatments led to internalized stigma and withdrawal from important aspects of life due to consequent disfigurement, including avoidance of sexual intercourse due to fears of exacerbating their condition. Anxieties related to possible disease progression sat alongside dissatisfaction with traditional treatments and fears that the treatments were ineffective or could be making things worse. Notably, only one person had been formally diagnosed and treated within the health system; most sought care from traditional healers.

The use of traditional treatments like *Euphorbia* plant latex and cauterization, leading to negative effects, is common among individuals with CL around the world [7,29,31]. *Euphorbia* species, such as *E. ingens* and *E. tirucalli*, are known for their antimicrobial properties and are used in some communities to treat skin conditions including wounds, abscesses, burns, ulcers, warts, skin tumors, and to remove tattoos; however, caution is advised due to the risk of skin irritation and toxicity from their latex, with some evidence linking the active agent to skin cancer [42,43]. There is little information on the pharmacological properties of other plants commonly used to treat CL in Ethiopia [19]. Studies from Suriname, Sri Lanka, and Ecuador indicated that most non-prescribed substances used by affected persons to treat CL at home were harmful [7,29,31]. These risks, alongside the fears of and dissatisfaction with traditional treatments we identified in Kalu underscores the need both to improve CL-affected people's access to safe and effective treatments through the health system as well as to share information on the adverse effects of commonly used home treatments.

We noted CL had a particularly marked impact on women's and younger persons' self-esteem, relationships, and emotional well-being consistent with studies of CL and other skin disorders affecting the face, highlighting that stigma experiences often intersect with other characteristics of social vulnerability like gender can contribute to worse effects in these groups [4,44–48].

While CL is a stigmatizing condition, some people in our study community perceived it differently, offering support rather than discrimination. Participants highlighted how the understanding that CL is non-contagious, the belief that anyone could be affected by the condition, and cultural attitudes that emphasize shared vulnerability and non-stigmatizing behaviors,

and changing social norms contribute to de-stigmatization. These findings underscore the importance of leveraging disease awareness and cultural narratives to develop stigma-reduction interventions.

Individuals with CL also experienced anxiety due to the perception of CL as a disfiguring disease, fearing disease progression and death from CL, as well as spreading it to others. As reported elsewhere, these fears contributed to the psychological impact [14,16,19]. Limitations in everyday activities such as dressing or getting out of bed reported in Tunisia [15] were not reported by our interviewees. However, we did find that many adults with CL reported abstaining from sex for fear that it would reduce the potency of traditional medicine or exacerbate their condition. We could find no evidence of other diseases of the skin being linked to sexual intercourse in this way in Ethiopia although studies from other settings report people with disorders such as psoriasis, eczema, acne, or lichen planus may avoid sexual intercourse due to physical and psychological factors associated with their skin condition affecting intimacy [49,50].

Our study has limitations. The majority of participants we recruited had presumed CL, which had not been parasitologically confirmed, raising the possibility that some of the lived experiences we have captured are not due to CL. This reflects challenges and barriers to care-seeking in the region, where individuals commonly must travel for several hours to access the only facility where diagnostic testing is available. The number of children we recruited was low, which may have limited our ability to explore the specific issues related to the lived experiences of children with CL. We could not fully investigate intersectional aspects such as any specific consequences for women or girls, and the young people who abstain from sex because of CL. The existence of structural stigma in the community's social services systems was not systematically probed in the narration of participants' experience. Comparing CL-related stigma with that of other conditions was not a focus of study. While acknowledging these limitations, we tried to address these limitations through transparent reporting, purposive sampling, and reflexive qualitative methods. Future studies should aim to address these gaps. We also did not systematically investigate all the plant medicines participants reported using and the names of many of the medicines given by healers were not known by the people being treated.

Nevertheless, our study has provided key insights into the lived experiences of people with CL in Ethiopia. We have demonstrated the significant physical, psychological and social impacts of the disease which emerge from the specific care-seeking and cultural context of Kalu but are also likely relevant to other endemic areas of Ethiopia. Our findings support calls by others that innovative interventions to increase access to treatment, reduce CL-related stigma and address other structural barriers to CL control are badly needed [51]. Providing information about the safety of common traditional treatments and correct information about contagiousness and how healing is affected by sexual intercourse are key areas that public health programmes could address to reduce some of the disease's impacts. Awareness about the curability and non-contagiousness of CL has been reported as a facilitator which helps reduce stigma in Sri Lanka [4]. Existing cultural attitudes that emphasize shared vulnerability to CL and underlie non-stigmatizing behaviors could also inform community-wide stigma interventions. More comprehensive approaches that centre CL-affected people's short- and long-term medical and psycho-social needs are also required. Identifying key aspects of stigma that contribute to ill mental health and poor quality of life for affected individuals is an important first step [4].

## Acknowledgments

We wish to thank the individuals and communities for their participation in the work of the Skin Health Africa Research Programme.

## Author contributions

**Conceptualization:** Maria Zuurmond, Saba Lambert, Katherine Halliday, Michael Marks, Mirgissa Kaba, Endalamaw Gadisa, Stephen L. Walker, Jennifer Palmer.

**Data curation:** Teklu Cherkose, Kibur Engdawork, Abebaw Y. Alemu, Fikregabrail Aberra Kassa, Zenebu Begna, Yohannes Hailemichael, Michael Marks, Mirgissa Kaba, Endalamaw Gadisa, Stephen L. Walker.

Formal analysis: Teklu Cherkose.

Funding acquisition: Saba Lambert, Michael Marks, Mirgissa Kaba, Endalamaw Gadisa, Stephen L. Walker, Jennifer Palmer.

Investigation: Teklu Cherkose, Kibur Engdawork, Yohannes Hailemichael, Michael Marks, Mirgissa Kaba, Endalamaw Gadisa, Stephen L. Walker, Jennifer Palmer.

Methodology: Teklu Cherkose, Kibur Engdawork, Maria Zuurmond, Tara Mtuy, Michael Marks, Mirgissa Kaba, Endalamaw Gadisa, Stephen L. Walker, Jennifer Palmer.

Project administration: Fikregabrail Aberra Kassa, Zenebu Begna, Tara Mtuy, Endalamaw Gadisa, Stephen L. Walker.

Resources: Kibur Engdawork, Maria Zuurmond, Saba Lambert, Michael Marks, Mirgissa Kaba, Endalamaw Gadisa, Stephen L. Walker, Jennifer Palmer.

Software: Fikregabrail Aberra Kassa, Stephen L. Walker.

Supervision: Kibur Engdawork, Maria Zuurmond, Saba Lambert, Tara Mtuy, Yohannes Hailemichael, Michael Marks, Mirgissa Kaba, Endalamaw Gadisa, Stephen L. Walker, Jennifer Palmer.

Validation: Kibur Engdawork, Maria Zuurmond, Fikregabrail Aberra Kassa, Saba Lambert, Tara Mtuy, Yohannes Hailemichael, Michael Marks, Mirgissa Kaba, Endalamaw Gadisa, Stephen L. Walker, Jennifer Palmer.

Visualization: Michael Marks, Stephen L. Walker, Jennifer Palmer.

Writing – original draft: Teklu Cherkose.

Writing – review & editing: Kibur Engdawork, Maria Zuurmond, Abebaw Y. Alemu, Fikregabrail Aberra Kassa, Saba Lambert, Zenebu Begna, Katherine Halliday, Tara Mtuy, Yohannes Hailemichael, Michael Marks, Mirgissa Kaba, Endalamaw Gadisa, Stephen L. Walker, Jennifer Palmer.

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
