## [Decision Letter · Decision Letter 0]

5 Nov 2025

“It left me burnt”: traditional treatment and stigma experiences of cutaneous leishmaniasis in Kalu district, Ethiopia

Dear Dr. Cherkose,

Thank you for submitting your manuscript to PLOS Neglected Tropical Diseases. After careful consideration, we feel that it has merit but does not fully meet PLOS Neglected Tropical Diseases's publication criteria as it currently stands. Therefore, we invite you to submit a revised version of the manuscript that addresses the points raised during the review process.

Please submit your revised manuscript within 60 days Jan 04 2026 11:59PM. If you will need more time than this to complete your revisions, please reply to this message or contact the journal office at plosntds@plos.org. Please include the following items when submitting your revised manuscript:

We look forward to receiving your revised manuscript.

Kind regards,

Christine A. Petersen

Academic Editor

Guilherme Werneck

Section Editor

Shaden Kamhawi

co-Editor-in-Chief

Paul Brindley

co-Editor-in-Chief

**Journal Requirements:**

At this stage, the following Authors/Authors require contributions: Kibur Engdawork, Maria Zuurmond, Abebaw Y. Alemu, Fikregabrail Aberra Kassa, Saba Lambert, Zenebu Begna, Katherine Halliday, Tara Mtuy, Yohannes Hailemichael, Michael Marks, Mirgissa Kaba, Endalamaw Gadisa, Stephen L. Walker, and Jennifer Palmer. Please ensure that the full contributions of each author are acknowledged in the "Add/Edit/Remove Authors" section of our submission form.

2) We note that you have indicated that there are restrictions to data sharing for this study. PLOS only allows data to be available upon request if there are legal or ethical restrictions on sharing data publicly. For more information on unacceptable data access restrictions, please see https://journals.plos.org/plosntds/s/data-availability#loc-unacceptable-data-access-restrictions.

b) If there are no restrictions, please upload the minimal anonymized data set necessary to replicate your study findings to a stable, public repository and provide us with the relevant URLs, DOIs, or accession numbers. For a list of recommended repositories, please see https://journals.plos.org/plosone/s/recommended-repositories. You also have the option of uploading the data as Supporting Information files, but we would recommend depositing data directly to a data repository if possible.

3) Please amend your detailed Financial Disclosure statement. This is published with the article. It must therefore be completed in full sentences and contain the exact wording you wish to be published.

1) State the initials, alongside each funding source, of each author to receive each grant. For example: "This work was supported by the National Institutes of Health (####### to AM; ###### to CJ) and the National Science Foundation (###### to AM).".

**Reviewers' Comments:**

Reviewer's Responses to Questions

**Key Review Criteria Required for Acceptance?**

**Methods**

-Are the objectives of the study clearly articulated with a clear testable hypothesis stated?

-Is the study design appropriate to address the stated objectives?

-Is the population clearly described and appropriate for the hypothesis being tested?

-Is the sample size sufficient to ensure adequate power to address the hypothesis being tested?

-Were correct statistical analysis used to support conclusions?

-Are there concerns about ethical or regulatory requirements being met?

Reviewer #1: The study provide a clear articulated and testable hypothesis, with a sound study design, however, there are several issues in the methods that need to be addressed. Specifically, the study has a small sample size and unclear selection mechanism reduce confidence that the sample represents the broader CL-infected population in Kalu. Further, the authors only recruited one confirmed CL participant, limiting the representativeness and subgroup interpretation. Additionally the study lacks a comparison group, thus, it is not possible to determine whether reported stigma and experiences are specific to CL rather than common to people with visible facial lesions or skin disease more broadly in the Kalu district. Finally the study only used a single coder thus lacking the ability to measure intercoder agreement or reliability. Thus, the authors cannot ensure coding consistency and credibility in qualitative research, and increase study bias. The authors must: fully describe recruitment, inclusion/exclusion criteria, and nonparticipant data. Additionally, they need include a comparator group and add additional coders to improve the credibility of the results.

Reviewer #2: The objective of the study, to gain information/evidence (rather than data I would argue) to support intervention design is clearly stated.

The study population is clearly identified and described. No power calculations were done as this very much a descriptive study of patient experience.

No ethical or regulatory concerns.

**Results**

-Does the analysis presented match the analysis plan?

-Are the results clearly and completely presented?

-Are the figures (Tables, Images) of sufficient quality for clarity?

Reviewer #1: The authors provide clear themes and some breakdown of the number of participants that responded within each theme, however, a demographic table of participants is needed. Additionally, please provide a flow chart of potential study participants and why/if they were excluded from the study. Finally, provide a table of all identified themes and how many participants responses fit into each theme.

Reviewer #2: Data analysis not applicable.

Descriptive results - I don't think there is any other way to present the report

No figures, tables etc

**Conclusions**

-Are the conclusions supported by the data presented?

-Are the limitations of analysis clearly described?

-Do the authors discuss how these data can be helpful to advance our understanding of the topic under study?

-Is public health relevance addressed?

Reviewer #1: The small sample size, lack of transparency surrounding participant selection, and single coder. limit the accuracy and generalizability of the study. thus, while the manuscript discusses the themes observed in the very small interview sample are as meaningful, general, or causal, they may actually reflect random variation. While some limitations are discussed further discussion surround how these limitations were overcome is needed, accounted for or will be accounted for is needed. Additionally, the limitations listed above need to be addressed.

Reviewer #2: Study limitations fully acknowledged.

The results are descriptive/narrative but are contextualised into stigma and skin disease milieu. There is no data as such and the number of participants seems on the low side but that is not to dismiss the qualitative information gathered.

**Editorial and Data Presentation Modifications?**

Reviewer #1: In line 76, instead of ‘is a skin neglected tropical disease (NTD)’, use ‘is a neglected tropical disease (NTD) of the skin’

In line 77, add a comma after ‘CL is characterized by ulcers, nodules’

In line 87, add a comma after ‘CL affected people experience stigma typified by rejection, isolation’

In line 104, add a comma after ‘in a specific socio-cultural context’

In line 131, the authors state, ‘People with presumed (n=17);’ however, throughout the manuscript, the authors do not make a note or acknowledge this. Rather, they use absolute language such as: ‘All interviewees had CL involving the face’. Please review the manuscript and add language to highlight that while all participants are suspected to have CL, they do not all have confirmed CL. For example, ‘all interviewees had CL-like or CL-presenting symptoms on their faces’.

In line 150, replace which in ‘we present themes which emerged’ with that

In line 158, replace which in ‘structural stigma which relates’ with that

In line 351, add a comma after ‘fearing disease progression and death from CL’

In line 355, add the after ‘fear that it would reduce’

In line 360, add a comma after ‘we recruited had presumed CL’

Reviewer #2: Accept. Think this is could be a short letter or article as it is descriptive/qualitative. There is no data to speak of.

**Summary and General Comments**

Reviewer #1: The manuscript ‘“It left me burnt”: traditional treatment and stigma experiences of cutaneous leishmaniasis in Kalu district, Ethiopia,’ by Cherkose et al., offers novel qualitative insight into the physical, emotional, and social effects of cutaneous leishmaniasis (CL) in residents of Kalu district, Ethiopia. Few studies have examined these factors and fewer still have looked at a Kalu district, Ethiopia specifically. Additionally, the examination of traditional medicine and the avoidance behavior is a novel addition. However, serious methodological shortcomings undermine confidence in the findings. The small sample size with poorly described recruitment methods introduces clear selection bias and limited generalizability. Further, the absence of any counterfactual or comparator group prevents the determination of whether reported practices and suffering are specific to CL. Finally, the manuscript lacks a transparent thematic analysis description and only uses a single coder, allowing selective reporting and a lack of intercoder reliability.

Revision: fully describe recruitment, inclusion/exclusion criteria, and nonparticipant data. Additionally, they need to report diagnostic status for each participant and either include a comparator group or reframe claims as exploratory and non-specific to CL. Additionally, the authors must present a reproducible analytic approach with a codebook, counts for each theme in a table format, and perform inter-coder procedures with the addition of a second coder and a tie breaker coder for disagreements.

Reviewer #2: I think this is a neat paper. I am familiar with CL as an infectious disease but have less knowledge about the stigma associated with skin NTDs and how that stigma manifests. My concerns are around the sample size and lack of parasitological confirmation but I do think the results show the importance of improved awareness around CL for those affected and for the communities involved.

It is a well written and presented manuscript clearly identifying the need to understand the existing context of CL in an area prior to designing a putative intervention.

PLOS authors have the option to publish the peer review history of their article (what does this mean? ). If published, this will include your full peer review and any attached files.

**Do you want your identity to be public for this peer review?** For information about this choice, including consent withdrawal, please see our Privacy Policy .

Reviewer #1: No

Reviewer #2: No

**Figure resubmission:**
---

## [Decision Letter · Decision Letter 1]

7 Feb 2026

Dear Mr. Cherkose,

We are pleased to inform you that your manuscript '“It left me burnt”: traditional treatment and stigma experiences of cutaneous leishmaniasis in Kalu district, Ethiopia' has been provisionally accepted for publication in PLOS Neglected Tropical Diseases.

Best regards,

Christine A. Petersen

Academic Editor

Guilherme Werneck

Section Editor

Shaden Kamhawi

co-Editor-in-Chief

Paul Brindley

co-Editor-in-Chief

Dear authors-

There are some small discrepancies in your demographics that should be corrected before publication. If you could please address the comments of reviewer 1 regarding # of males in the study this will suffice.

Reviewer's Responses to Questions

**Key Review Criteria Required for Acceptance?**

**Methods**

-Are the objectives of the study clearly articulated with a clear testable hypothesis stated?

-Is the study design appropriate to address the stated objectives?

-Is the population clearly described and appropriate for the hypothesis being tested?

-Is the sample size sufficient to ensure adequate power to address the hypothesis being tested?

-Were correct statistical analysis used to support conclusions?

-Are there concerns about ethical or regulatory requirements being met?

Reviewer #1: Sampling Method Limitations:

Although purposive sampling is an appropriate qualitative approach, it introduces potential selection bias that restricts generalizability. Because participants are chosen based on interviewer discretion, the resulting sample may not adequately represent broader populations. The additional use of snowball sampling could further compound this effect, as participants are likely to refer others with similar characteristics, thereby increasing homogeneity. These limitations must be clearly acknowledged and discussed within the manuscript.

Method for Confirming Agreement:

The manuscript mentions that agreement was confirmed, but provides little information on how this was achieved. The authors should briefly describe the method used to confirm agreement, including whether multiple analysts were involved and if a tie-breaking process was implemented. This information is essential to support the credibility and validity of the identified themes in the results.

Reviewer #3: I agree with Reviewer 2 that there may be methodological limitations; however, the information presented in this publication remains valuable to the public. Ideally, patients should have been clinically tested to confirm that they were indeed suffering from CL. That said, the authors have clearly stated that the objective of the study was to evaluate stigma associated with perceived CL infection rather than confirmed disease, which partially mitigates this limitation.

**Results**

-Does the analysis presented match the analysis plan?

-Are the results clearly and completely presented?

-Are the figures (Tables, Images) of sufficient quality for clarity?

Reviewer #1: Demographic Table Error:

In the manuscript, the authors state that "eleven of the 18 people interviewed were male"; however, the demographic table lists 13 males. This discrepancy should be corrected. Furthermore, the table should not include patient identification numbers. Instead, it should be structured with balanced variables that stratify participant groups, allowing readers and reviewers to accurately interpret the data distribution. Rather than listing each participant, the authors should present pooled counts stratified by the selected variables.

Reviewer #3: The paper primarily addresses perceptions and social conceptions surrounding CL. As such, the scope for rigorous scientific measurement and quantitative analysis is inherently limited, given the subjective nature of the information collected. While this constrains the depth of methodological analysis, it is consistent with the study’s stated objective of exploring stigma related to perceived CL infection.

**Conclusions**

-Are the conclusions supported by the data presented?

-Are the limitations of analysis clearly described?

-Do the authors discuss how these data can be helpful to advance our understanding of the topic under study?

-Is public health relevance addressed?

Reviewer #1: (No Response)

Reviewer #3: Given the above considerations, I agree with Reviewer 2 that this work has merit and could be valuable if published as a short article or a letter to the editor.

**Editorial and Data Presentation Modifications?**

Reviewer #1: (No Response)

Reviewer #3: (No Response)

**Summary and General Comments**

Reviewer #1: The revised manuscript demonstrates progress in addressing earlier methodological weaknesses, including acknowledgment of the limitation of presumed versus confirmed VL and clarification of recruitment and thematic analysis processes. However, several issues still limit confidence in the study’s rigor. The authors should clarify how agreement among analysts was confirmed during thematic coding and address the limitations introduced through the use of purposive sampling combined with snowball referrals. Additionally, the demographic data require correction, as the reported number of male participants differs between the text and the table. The table should exclude patient identifiers and instead present measures of central tendency in addition to counts stratified by key variables, placed appropriately within the Results section. Overall, while improvements are evident, the manuscript requires additional revisions before it can be considered for acceptance.

Reviewer #3: (No Response)

PLOS authors have the option to publish the peer review history of their article (what does this mean? ). If published, this will include your full peer review and any attached files.

**Do you want your identity to be public for this peer review?** For information about this choice, including consent withdrawal, please see our Privacy Policy .

Reviewer #1: No

Reviewer #3: No

---

## [Editor Report · Acceptance letter]

Dear Mr. Cherkose,

We are delighted to inform you that your manuscript, "“It left me burnt”: traditional treatment and stigma experiences of cutaneous leishmaniasis in Kalu district, Ethiopia," has been formally accepted for publication in PLOS Neglected Tropical Diseases.

Best regards,

Shaden Kamhawi

co-Editor-in-Chief

Paul Brindley

co-Editor-in-Chief
